# Gender-based violence care in Mauritania: Experience and caseload of six specialized hospital units (2018–2023)

**Clairanne Bost** [1] *, **Mouhamedou Diagana**[2], **Houssein Lebkem**[3]

1 Médicos del Mundo-España, Nouakchott, Mauritania, 2 Urology Department, Cheikh Zayed Hospital, Nouakchott Faculty of Medicine, Nouakchott, Mauritania, 3 Forensic Department, Military Hospital, Nouakchott, Mauritania

* coord.med.genre.mauritanie@medicosdelmundo.org

**Data Availability Statement:** Dataset and dataset description are uploaded as supplementary material to this research.

## Abstract

Since 2017, six specialized care units, the USPEC (*Unités Spéciales de Prise en Charge*) have been implemented in Mauritanian hospitals with the support of the international organization *Médicos del Mundo*. They provide healthcare and comprehensive assistance to victims of gender-based violence (GBV), such as sexual violence (SV), intimate-partner violence (IPV), female genital mutilation (FGM), adolescent pregnancy and child marriage. In this retrospective, observational study, we investigated the caseload of the six USPEC countrywide between January 1st, 2018, and June 30th, 2023. We analyzed consultation data, victims' sociodemographic characteristics, types of violence they were subjected to, specific patterns–location, relationship with the perpetrator, reoccurrence–, and medical care they received. 3550 cases were attended to, with a threefold increase in the mean number of monthly cases between 2018 and 2023. Women and girls accounted for 95.1% of victims; 78.7% were under 18 years old and 21.9% were under 12. All male victims (n = 172) were children. SV represented 79.8% of the caseload, early marriage/pregnancy 10.4%, IPV 7%, and FGM 0.7%. 80% of perpetrators were known to the victims, and the acts of violence had taken place in the victims´ own home for 60%. The proportion of cases received within 72 hours increased noticeably within the first two years before stabilizing at an average 81.3%. 7.21% of patients received local or surgical treatment and 1.8% were hospitalized. After SV, 996 received emergency contraception while 627, who sought care with delay, were already pregnant. Our findings suggest that the USPEC model responds to both victims' and the health system's needs to address GBV. Prevention, declaration and follow-up of pregnancy as a result of SV in young girls, likely constituted a major motivation for healthcare-seeking, yet more research is needed to document bottlenecks faced by GBV victims to access such services.

## Introduction

Over the last decades, increased awareness of the global magnitude of gender-based violence (GBV) has revealed a public health issue and human rights violations of alarming proportions

**Funding:** The authors received no specific funding for this work.

**Competing interests:** The authors have declared that no competing interests exist.

[1, 2]. Defined by the United Nations as "any act that results in, or is likely to result in, physical, sexual, or mental harm or suffering to women, including threats of such acts, coercion or arbitrary deprivation of liberty, whether occurring in public or in private life" [3], GBV encompasses domestic abuse, rape, sexual assault, sexual exploitation, so called "honor-based" violence, female genital mutilation and forced marriage. According to the World Health Organization (WHO), 30% of women worldwide have been subjected to such physical and/or sexual violence at some point in their lives [4]. A growing body of literature intends to quantify and qualify the phenomenon, thus defining its specific epidemiology [5–7]. In parallel, there is a need to contextualize such violence, not only in order to guide public policies–by assessing interventions aimed at preventing and mitigating its consequences [8–10]–, but also, as GBV patterns relate to social determinants. The ecological model of GBV, developed by L. Heise as early as 1998 [11], apprehends issues of power and domination, and suggests that gender discrimination at various levels of the society–micro, such as gender norms and stereotypes, or macro, such as laws and public policies–fuels the phenomenon of violence against women and children [1, 12].

The Islamic Republic of Mauritania is ranked 161st out of 170 by the United Nations Development Program's Gender Inequality Index [13]; the Social Institutions and Gender Index, which measures discrimination against women, ranked it last in 2021 [14]. Pervasive gender-based violence undoubtedly relates to this socio-cultural context of patent gender inequality that favors women's and girls' silence, subordination and limited access to support services. Islam plays a paramount role in Mauritania's social organization, notably through the application of sharia law. The Mauritanian penal code criminalizes rape (Art. 309) but also the act of *zina*, which qualifies sexual relationships outside marriage such as adultery (Art. 307). Article 307 specifies that the crime of *zina* is committed voluntarily by an adult, Muslim person, and hence excludes non-consensual intercourse and children. Still, well-documented gender discrimination in law enforcement exposes victims of Sexual Violence (SV) to the risk of prosecution and even detention for *zina*, regardless of a context of abuse and threat [15]. Such danger of double-victimization constitutes a major obstacle to victims disclosing SV they have been subjected to.

Yet much caution should be exercised before claiming a deterministic link between this context and a supposed effect on GBV, especially as pervasive discourse instrumentalizing the oppression of women by political Islam may skew interpretations [16]. At country-level, the pretense of religion to perpetrate violence against women is actively fought against, as is illustrated in the case against female genital mutilation (FGM). In Mauritania, FGM originated from the Peulh people, who inherited it from their Dogon neighbors [17]; although a religious justification is used by its defenders, the allegation of an Islamic origin of FGM cannot be sustained. Among the different ethnic groups in Mauritania, the Soninké and Haratine people name the practice with a religious connotation (*sallindé* and *sallu* respectively, both meaning prayers), whereas in the same milieu, the Wolof and Halpular have kept animist, pre-Islamic, names. Nowadays, whilst FGM is still illegally perpetuated, The Mauritanian General Child Protection Code describes it as a cruel, inhumane or degrading treatment, punishable by 1 to 3 years' imprisonment and a fine. Additionally, 32 Mauritanian authoritative ulemas and imams signed a Fatwa in 2010 to prohibit FGM, considered an anti-Islamic practice since the religion condemns any act detrimental to health.

Mauritania thus deserves special investigation; however, neither the WHO databases [4, 18] nor the scientific literature [5, 6, 12] document the patterns and scale of GBV in the country. A review of the literature reveals only one article for Mauritania [19], which describes two specific forms of violence against women and girls: force-feeding and FGM. This study, based on national health data 2000–2001, indicates an FGM prevalence rate of 77%, along with largely

favorable attitudes in the population towards the practice. Mauritania has since engaged in a bold process of documenting and eradicating GBV. In 2011, the *Enquête Nationale sur la Violence à l'Égard des Femmes en Mauritanie* (National Survey on Violence against Women in Mauritania) [20] broadens the scope of the forms of violence investigated to include physical, sexual, psychological and economic violence, as well as violence linked to the non-application of the law. Considering the first three categories, it establishes an overall prevalence rate of GBV of 68.1% among the adult female population surveyed–much higher than global statistics, around 23% to 30% of the adult female population [4]. In 2022, it is complemented by the *Enquête Démographique et de Santé en Mauritanie 2019–2021* [21], hereafter "EDSM 2021", which documents the prevalence of FGM (64% nationally and up to 94% in some regions), although attitudes in favor of the practice have seemingly declined– 38% of women and 48% of men surveyed. It also records the proportion of pregnant adolescents or those who have given birth to their first child, which rises from 7% at age 15 to 31% at age 19 and, presumably intertwined with the context of social and legal norms forbidding pregnancy outside wedlock, child marriage: 17% of girls under 15 years old and 39% of girls under 18 years old were already married. Still, no studies record the impact of such violence on the physical and mental health of victims in Mauritania, and certain types of violence, arguably those less visible–e.g. economic, such as the denial of resources by spouses; reproductive, such as the lack of access to family planning for unmarried women; trafficking and exploitation–remain undocumented.

Upholding the human rights and health rights that GBV denies, demands to strengthen the healthcare system to provide supportive care [22, 23], prevent violence [2, 10], and fuel evidence-based advocacy [24]. The provision of robust data from specialized GBV care units constitutes a relevant opportunity to raise awareness about the phenomenon [25]. Complementing national statistics, such data demonstrates the health impact of violence, and paves the way to improve care and services for those who agree to disclose the violence they have been subjected to. In this study, we sought to understand the scope, patterns and health outcomes of GBV in Mauritania, and how they relate to the country's socio-legal context, by examining a six-year caseload of victims seeking medical attention in GBV-specialized hospital units. We further aimed to demonstrate that these units exemplify an advanced, comprehensive care model that effectively meets the health needs of the population.

## Materials and methods

### Study design

In this retrospective observational study, we investigated the caseload of GBV victims who received care in six specialized, tertiary-level units countrywide, between January 1st, 2018, and June 30th, 2023. Based on secondary data routinely collected in patients' files, we examined the victims' demographic characteristics, the types and patterns of violence they were subjected to, and the medical care they received.

### Study setting

Since 2017, a model of specialized care units, the USPEC (*Unités Spéciales de Prise en Charge [des victimes de violence de genre]*) has been progressively implemented in Mauritania with the support of *Médicos del Mundo* (MdM) Spain, an international non-governmental medical organization committed to a health-as-a-right approach, with special focus on sexual and reproductive health.

The USPEC model consists in an in-hospital unit set up within the maternity ward, open 24/7, and staffed with one to two dedicated midwives or nurses, psychosocial assistants and

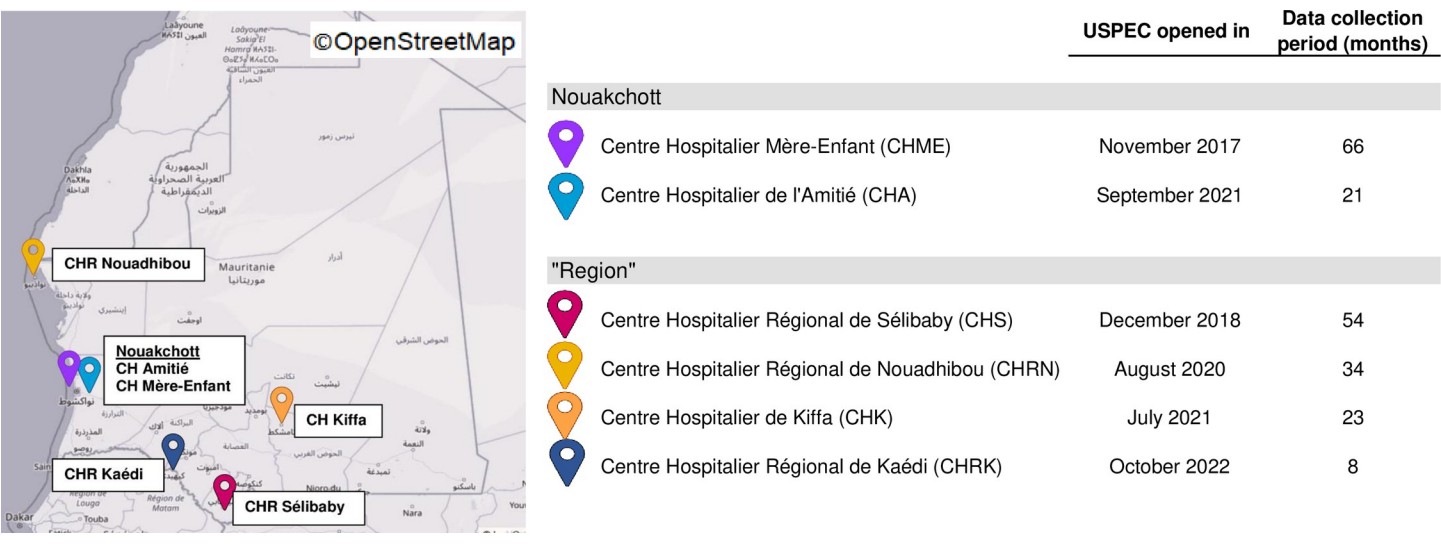

**Fig 1. Location of the first 6 USPEC opened in Mauritania.**

social workers. Alongside on-call physicians–gynecologists, pediatricians, forensic specialists–, healthcare professionals provide free, emergency and delayed medical attention according to WHO guidelines: physical examination, laboratory testing and additional investigations (e.g. x-ray, ultrasound, etc.), treatment provision (post-exposure prophylaxis, wound care, etc.), delivery of a medico-legal certificate upon request, and appropriate referral for specialized care. Trained psychosocial assistants provide initial and follow-up care, home visits and support group sessions. An essential part of the USPEC multidisciplinary teams, social workers are employed by two national organizations dedicated to advancing women's rights–*l'Association des Femmes Cheffes de Famille* and *l'Association Mauritanienne pour la Santé de la Mère et de l'Enfant*. They address victims' protection needs (emergency housing, income-generating activities, etc.) through internal referral, ensure the gratuity of all care received and transport utilized, and support victims throughout their contact with the police and justice system.

The first USPEC was opened in 2017 in a private hospital (*Centre Hospitalier Mère-Enfant, or CHME*) of the capital, Nouakchott. In the following years, another five USPEC have been progressively set up in regional hospitals of the most densely populated areas with the support of MdM: in Sélibaby from 2018, in Nouadhibou from 2020, in Kiffa and a second USPEC in Nouakchott (*Centre Hospitalier de l'Amitié, or CHA*) from 2021, and finally Kaedi, from 2022 (see Fig 1).

## Inclusion criteria

All cases involving GBV and receiving initial care (medical, psychosocial or otherwise) in one of the six USPEC supported by MdM between January 1st, 2018, and June 30th, 2023 were included. There were no age or sex restriction. GBV included, but was not limited to: Sexual Violence (SV), Female Genital Mutilation (FGM), Intimate Partner Violence (IPV), other physical / domestic violence, adolescent pregnancy and child marriage (or risk thereof), and reproductive coercion (see detail and definitions in Fig 2).

**Non-inclusion criteria.** All follow-up care cases; patients who presented to the USPEC but whose case did not involve GBV, or who did not receive care (e.g. looking for information, or redirected to another unit); victims who received care in other healthcare facilities.

Owing to the wide variety of gender-based violence types and the comprehensive attention the USPEC aim to deliver, the USPEC admission criteria are purposefully left non-binding, but generally encompass the categories listed below. The USPEC also welcome victims of psychological, vicarious, and economical violence to benefit from psychosocial support, as well as mediation procedures and/or judicial support through internal reference by the social workers within their organizations. Though GBV victims are often subjected to different forms and/or a continuum of violence, patients' files record a chief reason for consultation. For the purpose of this study, these were defined as follows:

**Sexual violence (SV)** was defined as any non-consensual, forced or coerced sexual contact, and sub-categorized as: *rape*, involving the penetration of the vagina, mouth or anus, using a penis, other body parts or an object; *collective rape*, when rape involved more than one perpetrator; *other non-penetrative sexual violence*, for attempted rape or any other sexual contact (touching) or abuse (brutality) that does not imply the risk of pregnancy or infection transmission associated with penetrative intercourse.

**Female Genital Mutilation (FGM)** was defined as "any procedure involving partial or total removal of the external female genitalia or other injury to the female genital organs whether for cultural or other non-therapeutic reasons" (World Health Organization).

**Intimate Partner Violence (IPV)** was defined as sexual, physical, economical, psychological or emotional violence, threat, and coercion, as well as non-consensual sexual intercourse, by a current or former intimate partner. IPV generally encompasses a continuum of these forms of violence but, owing to the healthcare-oriented nature of the USPEC, this category was subdivided in *physical violence* (i.e., requiring medical and medico-legal attention) and *other* (i.e. requiring psychosocial and judicial support, generally through the referral system).

**Other physical violence** encompassed kidnapping without associated sexual violence, and non-IPV domestic violence – for instance, perpetrated by parents against their child as a punishment for alleged sexual misconduct (e.g. spending the night outside), or between co-spouses in the context of polygamous unions (e.g. as a result of jealousy).

**Adolescent pregnancy and child marriage (or risk thereof)** was developed as a context-specific typology, distinct from SV, and stemming from another case definition that had spontaneously arisen amongst midwives: "*suspicion*". This category used to designate cases of adolescent girls brought by their parents, the police or both, "suspected" of having had sexual intercourse – to which, the medical or psychological interview revealed, they had consented, often with their same-age boyfriends. Hence, they were not strictly victims of "sexual violence" (though consent is not recognized before the age of 18 in Mauritania), and they were often brought to the USPEC out of genuine concern for the prejudice associated with the loss of virginity of an unmarried girl. These adolescents are welcomed to the USPEC, as they are potentially subjected to other forms of GBV that can be alleviated: for instance, a prosecution for *zina* which can result in a prison sentence even in an under-age child, or physical and economical violence from the parents and family as punishment (e.g. rejection from family home). In case of pregnancy, they also face a high risks of forced marriage as a family arrangement to "save face", school expulsion, loss of economic means, and maternal mortality and morbidity.

In the context of forbidden extra-marital sexual intercourse, the USPEC used to admit, under the similar label of "suspicion", patients over 18 years brought by the police with a requisition forcing them to undergo a gynecological examination, as part of a prosecution for *zina*. The cases, here categorized as "**Reproductive coercion**" are no longer admitted, as the USPEC have gained traction and are now able to refuse to such cases. Only at the victim's demand, and provided she was subjected to another form of violence (and as per her declaration), can she be admitted.

**Other forms of violence** included economical, psychological and emotional violence.

**Fig 2. Types of gender-based violence attended to in the USPEC.**

**Exclusion criteria.**   Cases with insufficiently detailed records–at least a consultation date and place had to be filled in.

## Data collection and material

Since the opening of the first USPEC in 2017, data from patients' file has been routinely collected and anonymously aggregated in an excel spreadsheet filled by USPEC supervisors employed by MdM, on a weekly basis. For this study, we investigated the following parameters extracted from the database:

Consultation characteristics: date, location (USPEC), delay between GBV and consultation;

Patients' characteristics: sex, age group, residence, education level, and marital status;

GBV characteristics: type, location, relationship with the perpetrator, reoccurrence with the same perpetrator;

Medical care received: local or surgical treatment following lesions; hospital admissions and length of stay, unintended pregnancy diagnosed or prevented with emergency contraception.

## Data access and analysis

In this study, data were accessed between July and September 2023 for data cleaning and database creation. From September to December 2023, the database was re-encoded into Python and accessed for analysis and visualization purpose. The scope of this research and available resources dictated a straightforward approach to data analysis: with a focus on gathering baseline information and identifying potential patterns, along with a high number of parameters investigated, and the categorical and ordinal nature of the data collected, only descriptive statistics–frequencies, percentages, and proportions–were used to summarize and present them.

## Ethics statement

All patients involved in this study received thorough information at their admission in the USPEC. Owing to the sensitive nature of GBV, the USPEC team actively seek their informed consent, and notify them of their right to refuse all or part of the treatment (examination, collection of evidence), as well as their right to lodge a complaint or not. Children and minors are always accompanied by a parent, tutor or responsible adult: both are informed according to their discernment capacity. Information and consent to healthcare are currently reported on patients' files and signed by social workers.

Only routine data routinely collected for care purposes, independently of the study, was investigated, with no additional consultation, investigation or biological material collection or involvement from patients. All data collected was aggregated by the USPEC supervisors'– nurses and midwives–in a password-protected database. Hospital administration are informed about which professional have access to patient records and database relating to all their patients, past and present. No protected health information appears in the database: file numbers, age group and city or region replace the name, birthdate and address that appear in patients' files.

As a non-interventional, retrospective study performed using available, de-identified data, that had been routinely collected for clinical purpose, it did not require individual consent specific to the research. For publication, researchers requested the approbation of an ethics

committee that this study complies with the regulations and reference methodologies for scientific integrity and the protection of participants. They obtained approval from the Mauritanian Ethics Advisory Board, and an ethical letter from MdM, signed by its legal representative, indicating that the research work has been carried out following the guidelines of the Declaration of Helsinki.

## Results

3550 cases were included in the present study, and 2 were excluded for a lack of sufficient data (only sex and age group were entered, and no consultation date nor file number allowed researchers to retrieve these records).

### Caseload and distribution

Fig 3 shows the caseload distribution per trimester and per USPEC over the period studied. All the USPEC started receiving cases either directly from their inauguration (see inauguration month in Fig 1) or in the following month. Between 2018 and 2023, the mean number of monthly cases tripled, from 22 to 66. In Nouakchott, the caseload of CHME rapidly increased over the first 18 months, reaching 60 cases per month mid-2020, and stabilizing at an average of 39.5 since. After a second USPEC opened in Nouakchott in 2021, this caseload has plateaued, though spread between CHA and CHME. With little to no increase overtime, the USPEC in Nouadhibou–the second-largest city in Mauritania–received an average of 7.4 cases per month, and the ones based in Sélibaby, Kiffa and Kaédi–the most rural regions–respectively 1.5, 2.6 and 1.4 cases per month.

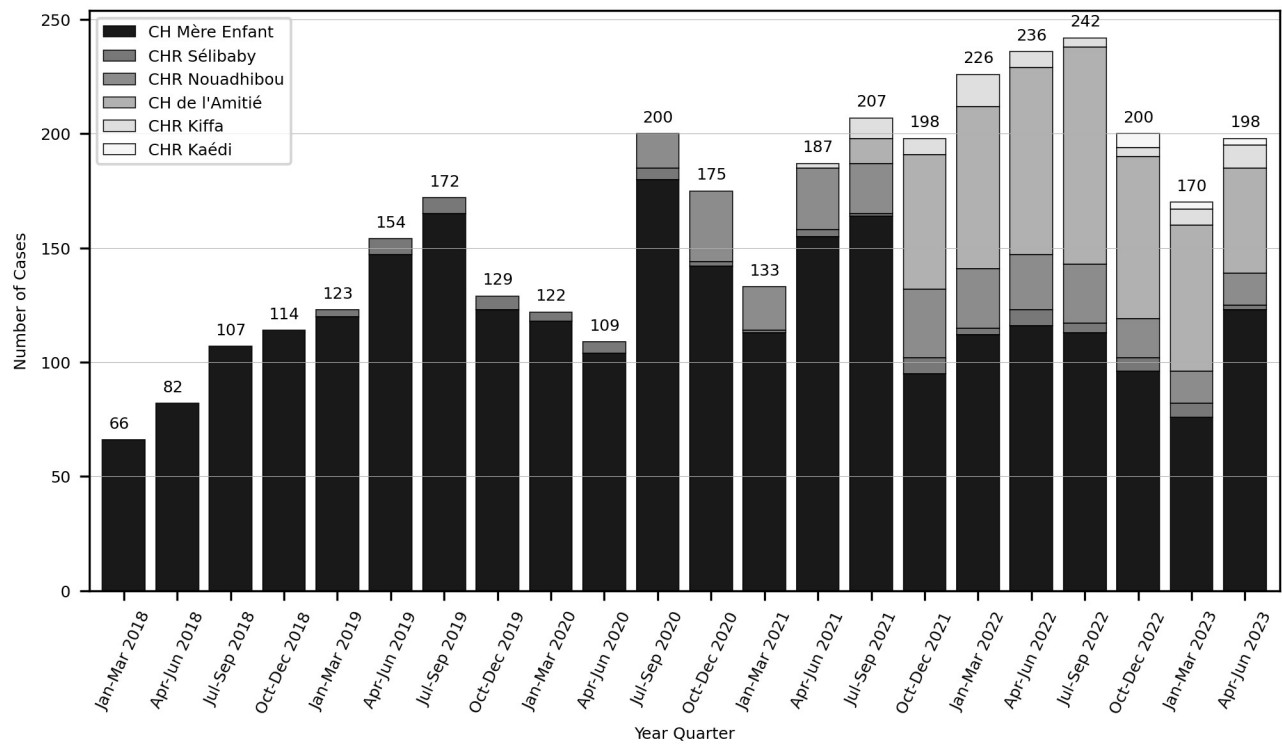

**Fig 3. Caseload distribution by USPEC over time.**

**Table 1. Victims demographic characteristics.**

| | Total USPEC | | General Population* |
|---|---|---|---|
| | Number | (%) | (%) |
| | 3550 | | |
| **Sex** | | | |
| Female | 3375 | 95.07% | 51.00% |
| Male | 172 | 4.85% | 49.00% |
| Missing values | 3 | | |
| **Age group** | | | |
| 0–5 | 183 | 5.15% | |
| 6–11 | 595 | 16.76% | |
| 12–17 | 2008 | 56.56% | |
| 18+ | 756 | 21.30% | |
| Missing values | 8 | | |
| **Residence** | | | |
| Urban: Total | 3483 | 98.11% | 48.30% |
| Nouakchott | 3110 | 87.61% | 27.10% |
| Nouadhibou | 255 | 7.18% | |
| Other | 118 | 3.32% | |
| Rural | 52 | 1.46% | 51.70% |
| Outside Mauritania | 2 | 0.06% | |
| Missing values | 13 | | |
| **Education level** | | | |
| None | 267 | 10.17% | 10.00% |
| Quranic school | 700 | 26.67% | 29.10% |
| Primary | 1195 | 45.52% | 39.00% |
| Secondary | 458 | 17.45% | 16.60% |
| Post-graduate | 5 | 0.19% | 3.30% |
| Missing values | 925 | | |
| **Marital status** | | | |
| Single | 581 | 50.88% | 26.60% |
| Married | 508 | 44.48% | 62.50% |
| Polygamous | | | 8.00% |
| Divorced | 52 | 4.55% | 9.40% |
| Widowed | 1 | 0.09% | 1.50% |
| Missing values | 2408 | | |

[a]Source: EDSM 2021. Marital status was considered among the 15–49 y women subgroup only.

## Demographic characteristics of GBV victims

As shown in Table 1, women and girls accounted for over 95% of victims. Nearly 78.7% were under 18 years old, with a striking 21.9% under 12. A small proportion of male victims were also attended to (n = 172), all of them under 18 years old. With 88.1% of cases consulted in Nouakchott, urban residence is most represented (98.1%). Owing to the high number of missing data (which were excluded in percentage calculation) and the high proportion of children in the cohort, neither education level nor marital status could be associated with significant trends. Still, even though it is not matched with age, the distribution in victim's education level does not differ from the general population as extensively as sex.

**Table 2. Types of GBV.**

| | Total USPEC | | Nouakchott | | Other regions | | Male victims | | With police requisition | |
|---|---|---|---|---|---|---|---|---|---|---|
| | **Number** | **(%)** | **Number** | **(%)** | **Number** | **(%)** | **Number** | **(%)** | **Number** | **(%)** |
| | 3550 | *100.00%* | 3130 | *100.00%* | 420 | *100.00%* | 172 | *100.00%* | 3420 | *100.00%* |
| **Sexual violence (SV)** | **2834** | ***79.83%*** | **2561** | ***81.82%*** | **273** | ***65.00%*** | **168** | ***97.67%*** | **2746** | ***80.29%*** |
| Rape | 1657 | *46.68%* | 1474 | *47.09%* | 183 | *43.57%* | 129 | *75.00%* | 1589 | *46.46%* |
| Collective rape | 185 | *5.21%* | 161 | *5.14%* | 24 | *5.71%* | 16 | *9.30%* | 177 | *5.18%* |
| Other, non-penetrative sexual violence | 992 | *27.94%* | 926 | *29.58%* | 66 | *15.71%* | 23 | *13.37%* | 980 | *28.65%* |
| **Female Genital Mutilation (FGM)** | **24** | ***0.68%*** | **12** | ***0.38%*** | **12** | ***2.86%*** | **0** | ***0.00%*** | **11** | ***0.32%*** |
| **Intimate Partner Violence (IPV)** | **248** | ***6.99%*** | **183** | ***5.85%*** | **65** | ***15.48%*** | **1** | ***0.58%*** | **236** | ***6.90%*** |
| Physical | 245 | *6.90%* | 181 | *5.78%* | 64 | *15.24%* | 1 | *0.58%* | 233 | *6.81%* |
| Other: psychological, economical | 3 | *0.08%* | 2 | *0.06%* | 1 | *0.24%* | 0 | *0.00%* | 3 | *0.09%* |
| **Adolescent pregnancy / child marriage** | **368** | ***10.37%*** | **310** | ***9.90%*** | **58** | ***13.81%*** | **2** | ***1.16%*** | **358** | ***10.47%*** |
| Marriage (or risk thereof) | 56 | *1.58%* | 50 | *1.60%* | 6 | *1.43%* | 2 | *1.16%* | 54 | *1.58%* |
| Pregnancy (or risk thereof) | 101 | *2.85%* | 89 | *2.84%* | 12 | *2.86%* | 0 | *0.00%* | 99 | *2.89%* |
| Other | 211 | *5.94%* | 171 | *5.46%* | 40 | *9.52%* | 0 | *0.00%* | 205 | *5.99%* |
| **Other physical/domestic violence** | **9** | ***0.25%*** | **7** | ***0.22%*** | **2** | ***0.48%*** | **0** | ***0.00%*** | **8** | ***0.23%*** |
| **Other** | **52** | ***1.46%*** | **43** | ***1.37%*** | **9** | ***2.14%*** | **0** | ***0.00%*** | **48** | ***1.40%*** |
| Reproductive coercion | 50 | *1.41%* | 42 | *1.34%* | 8 | *1.90%* | | | 48 | *1.40%* |
| Other | 2 | *0.06%* | 1 | *0.03%* | 1 | *0.24%* | | | 0 | *0.00%* |
| Missing data | 15 | | 14 | | 1 | | 1 | | 13 | |

## Typology of GBV cases

Sexual Violence (SV) accounted for 79.8% of the caseload (n = 2834 cases), with rape the main reason for consultation (n = 1842). Table 2 suggests some disparities in compared proportions of GBV types between the capital and the regions: SV represented over 81.8% of cases in Nouakchott, against 65% in the regions; IPV made up 7% of cases in Nouakchott against 15% in the regions. Though based on a very limited number (n = 24), cases of FGM were evenly distributed between Nouakchott and the regions. Adolescent pregnancy and child marriage (or risk thereof) represented 10.3% of cases–slightly more in the regions. 3420 victims (96,3%) presented with a police requisition, with no significant difference in police involvement depending on the type of GBV, apart from FGM (n = 11, out of 24 cases). 97.6% of boys had been subjected to sexual violence; only a small number had been subjected to other forms of GBV, such as IPV (n = 1) and forced marriage (n = 2).

## GBV patterns

In addition to the type of violence, specific patterns were investigated, such as the relationship between the victim and perpetrator, the reoccurrence of GBV by the same perpetrator, and the place where the act was committed (Table 3). These figures patently expose that more than 80% of abusers lived in the intimate sphere of the victims, and that the acts of violence had taken place in the victim´s home 60% of the time. In Nouakchott, where the proportion of IPV was lower than in regions, known perpetrators were generally not the intimate partners (11.1%) but relatives (45.5%)–fathers, uncles, grand-fathers, cousins–and the male entourage (27%)–neighbors and acquaintances, known shopkeepers, teachers, taxi-drivers. Unlike in rural areas where over 1 in 5 victims were taken to an unknown location (often designated as "the bush" in their statements), in Nouakchott 70% of them were subjected to violence in a

**Table 3. Patterns of GBV.**

| | Total USPEC | | Nouakchott | | Other regions | | Male victims | | With police requisition | |
|---|---|---|---|---|---|---|---|---|---|---|
| | Number | (%) | Number | (%) | Number | (%) | Number | (%) | Number | (%) |
| | 3550 | *100.00%* | 3130 | *100.00%* | 420 | *100.00%* | 172 | *100.00%* | 3420 | *100.00%* |
| *Perpetrator* | | | | | | | | | | |
| **Known perpetrator(s)** | **2966** | **83.55%** | **2636** | **84.22%** | **330** | **78.57%** | **156** | **90.70%** | **2874** | **84.04%** |
| Intimate Partner | 394 | *11.10%* | 292 | *9.33%* | 102 | *24.29%* | 3 | *1.74%* | 368 | *10.76%* |
| Relative: father, uncle, cousin, etc. | 1615 | *45.49%* | 1479 | *47.25%* | 136 | *32.38%* | 99 | *57.56%* | 1580 | *46.20%* |
| Other: friend, neighbor, acquaintance | 957 | *26.96%* | 865 | *27.64%* | 92 | *21.90%* | 54 | *31.40%* | 926 | *27.08%* |
| **Unknown perpetrator(s)** | **351** | **9.89%** | **308** | **9.84%** | **43** | **10.24%** | **12** | **6.98%** | **334** | **9.77%** |
| Missing data | 233 | *6.56%* | 186 | *5.94%* | 47 | *11.19%* | 4 | *2.33%* | 212 | *6.20%* |
| *Reoccurrence with the same perpetrator* | | | | | | | | | | |
| Yes | 615 | *17.32%* | 523 | *16.71%* | 92 | *21.90%* | 38 | *22.09%* | 584 | *17.08%* |
| No | 2912 | *82.03%* | 2596 | *82.94%* | 316 | *75.24%* | 134 | *77.91%* | 2819 | *82.43%* |
| Missing data | 23 | | 11 | | 12 | | 0 | | 0.17 | |
| *Location* | | | | | | | | | | |
| **Private location** | **2406** | **67.77%** | **2193** | **70.06%** | **213** | **50.71%** | **109** | **63.37%** | **2342** | **68.48%** |
| Victim's home | 2180 | *61.41%* | 2019 | *64.50%* | 161 | *38.33%* | 94 | *54.65%* | 2129 | *62.25%* |
| Other, including perpetrator's home | 226 | *6.37%* | 174 | *5.56%* | 52 | *12.38%* | 15 | *8.72%* | 213 | *6.23%* |
| **Public location** | **841** | **23.69%** | **653** | **20.86%** | **188** | **44.76%** | **57** | **33.14%** | **792** | **23.16%** |
| Taxi, transport | 117 | *3.30%* | 102 | *3.26%* | 15 | *3.57%* | 4 | *2.33%* | 112 | *3.27%* |
| School | 15 | *0.42%* | 12 | *0.38%* | 3 | *0.71%* | 5 | *2.91%* | 15 | *0.44%* |
| Other: street, shop, etc. | 566 | *15.94%* | 486 | *15.53%* | 80 | *19.05%* | 38 | *22.09%* | 539 | *15.76%* |
| Unknown by the victim | 143 | *4.03%* | 53 | *1.69%* | 90 | *21.43%* | 10 | *5.81%* | 126 | *3.68%* |
| Missing data | 303 | *8.54%* | 284 | *9.07%* | 19 | *4.52%* | 6 | *3.49%* | 286 | *8.36%* |

house: theirs, their abuser's or, less commonly, a rented place. 90% of male victims knew their abusers, and boys also had a higher percentage of reoccurrence (22.1%, compared to 17.32% amongst the total USPEC cohort) and were more likely to be raped at school (2.9%, compared to 0.42% amongst the total USPEC cohort).

## Consultation delay

As shown in Fig 4, the proportion of cases received within the delay of 72 hours after GBV increased noticeably within the first two years: from 12.1% in early 2018, it culminated above 99% in 2021 before dropping slightly and stabilizing, for the last 18 months, at an average of 81.3%.

## Medical care received

Table 4 provides a snapshot of the medical conditions that victims sought medical attention for. 7.2% received local–wound dressing, casting, suturing–or surgical treatment, a proportion higher after IPV (21%) or FGM (41.7%) than after SV (5.8%). Less than 2% were hospitalized, with a similar trend of increased needs after IPV (4.6%) or FGM (20.8%). Unintended pregnancy and risk thereof constituted the most common detrimental health outcome, especially considering that 80% of victims were under 18 (see Table 2): of 3550 cases, nearly a thousand were at risk of becoming pregnant–based on puberty scale assessment, penetration, existence

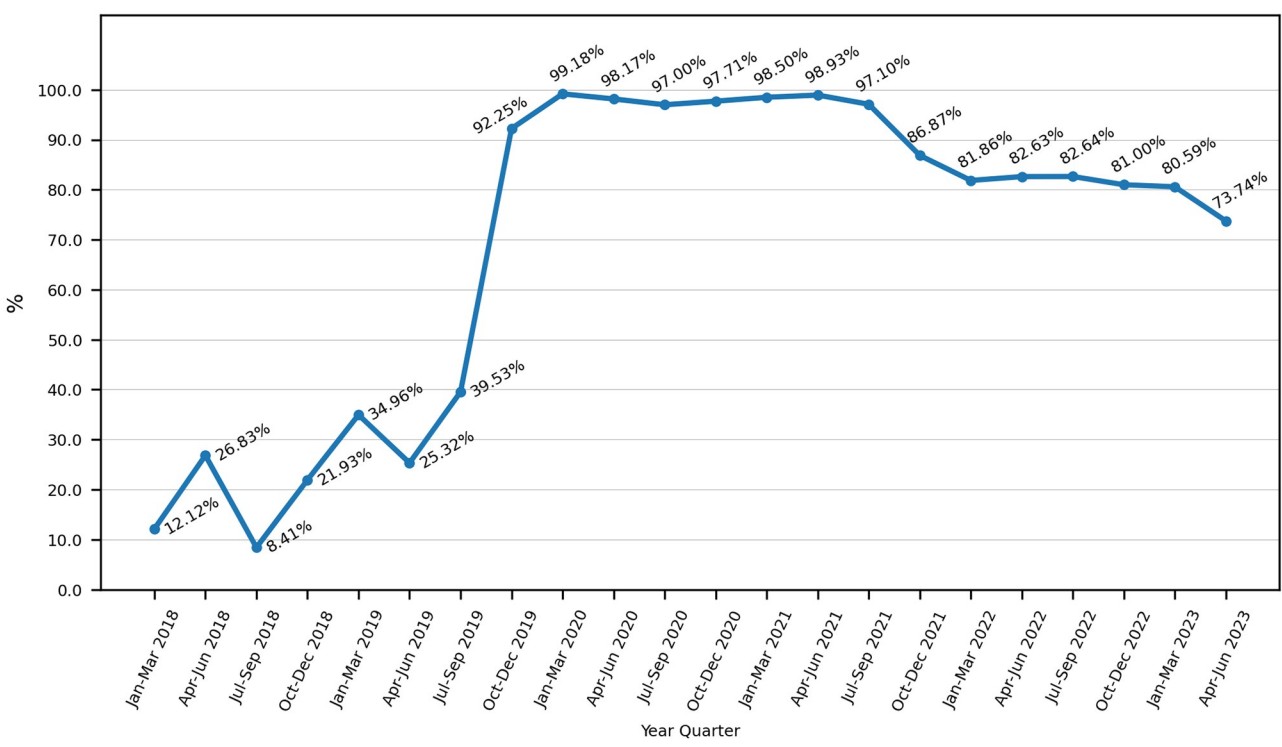

**Fig 4. Evolution of the percentage of cases consulting within 72 hours after GBV.**

**Table 4. Health outcomes.**

| | Total USPEC | | After SV (N = 2834) or adolescent pregnancy/ child marriage or risk thereof (N = 368) | | After IPV (N = 248) or non-IPV physical/domestic violence (N = 9) | | After FGM | |
|---|---|---|---|---|---|---|---|---|
| | Number | % Total | Number | % SV | Number | %IPV/other | Number | % FGM |
| | 3550 | | 3202 | | 257 | | 24 | |
| **Treatment for lesions** | **256** | *7.21%* | **188** | *5.87%* | **54** | *21.01%* | **10** | *41.67%* |
| Local treatment | 169 | *4.76%* | 117 | *3.65%* | 44 | *17.12%* | 7 | *29.17%* |
| Surgical treatment | 87 | *2.45%* | 71 | *2.22%* | 10 | *3.89%* | 3 | *12.50%* |
| **Hospitalization** | **64** | *1.80%* | **43** | *1.34%* | **12** | *4.67%* | **5** | *20.83%* |
| 1 day | 34 | *0.96%* | 23 | *0.72%* | 6 | *2.33%* | 2 | *8.33%* |
| 2+ days | 27 | *0.76%* | 19 | *0.59%* | 6 | *2.33%* | 2 | *8.33%* |
| Missing data | 3 | | 1 | | 0 | | 1 | |
| **Unintended pregnancy** | **1623** | *45.72%* | **1539** | *48.06%* | **59** | *22.96%* | **1** | *4.17%* |
| Emergency contraception delivery | 996 | *28.06%* | 975 | *30.45%* | 3 | *1.17%* | | |
| Confirmation of pregnancy (βHCG or ultrasound) | 627 | *17.66%* | 564 | *17.61%* | 56 | *21.79%* | | |

of a contraception means, existence of an earlier pregnancy–and received an emergency contraception. 627 were already pregnant due to sexual violence.

## Discussion

With 3550 cases consulted and a threefold increase in the mean monthly number of new cases consulted over the last 5 years, along with consultation starting immediately after each

inauguration of a new unit countrywide, the USPEC model seems to adequately respond to previously unaddressed health needs of the Mauritanian population. Low at first, the proportion of patients presenting within 72 hours after GBV has rapidly increased during the first two years of the USPEC implementation. This should be interpreted as a result of the awareness-raising interventions led by MdM in schools, communities and healthcare facilities. Rather than presenting spontaneously, the wide majority of victims were referred by the police. This suggests a success in concurrent training programs of police officers, but mostly demonstrates the paramount role of social workers deployed within the police brigades, the USPEC and law courts by civil society organizations to offer a safe and dignified support to victims of human rights violations. Their considerable work of orientation, protection and follow-up, both upstream and downstream of the USPEC, cannot be understated; their role as an interface between the hospital and the outside world constitutes a crucial component of the model. Aside from this main referral pathway, referrals from health centers and community health workers appeared lower, highlighting a need to train healthcare professionals on GBV identification. Improved screening and referral, especially during antenatal and postnatal care–which constitute key entry points for women within the health system–and for pregnant adolescents in the most vulnerable socio-economic situations, are likely the reasons for the observed drop in victims presenting after 72 hours. Still, most interventions delivered after SV, IPV and FGM are time-sensitive–medico-surgical emergency care, post-exposure prophylaxis after SV, medico-legal evidence collection, psychological first aid–and strategies to improve urgent referral, from public awareness campaigns to specific training for healthcare professional, are currently deployed by MdM, the Ministry of Health, and many other national and international organizations.

In this study, sexual violence was the most pervasive form of GBV attended to, accounting for nearly 80% of cases, and 78% of victims were girls under 18 years old. Our findings unambiguously refute any victim-blaming discourse: among thousands of victims, most were children, assaulted at home by someone they knew, and possibly trusted–such as a relative. Within GBV cases, a triad of "female victim under 18 years old—known perpetrator—related pregnancy", emerged, for which two concurrent hypotheses can be proposed. Firstly, this triad may be specific to, or enforced by, the Mauritanian context, i.e. the criminalization of intimate relationships outside wedlock and of abortion care, social worth attributed to virginity before marriage, and a lack of access to contraception for unmarried women. Secondly, the prevention, declaration and follow-up of pregnancy, especially adolescent pregnancy, may constitute a major motivation for healthcare-seeking and justice-seeking behavior among GBV victims. As documented in the EDSM 2021, adolescent pregnancy and child marriage are intertwined and both are pervasive in Mauritania. Perhaps more than other GBV victims, pregnant teenagers and their families keenly rely on the free, multidisciplinary care the USPEC offer, because it alleviates the economic burden associated with healthcare they face, and mitigates the risk associated with early pregnancy, both medical (maternal mortality and morbidity), and social (loss of economic means, or rejection from their environment, school and family). The aforementioned focus on strengthening referral pathways to the USPEC for the most vulnerable teenagers attending antenatal care could contribute to the national and international imperative to tackle maternal mortality and morbidity. Moreover, this is an opportunity for the USPEC to be considered less a "rape office [*le bureau des viols*]", as often heard in hospital corridors, and more as multidisciplinary units welcoming different situations, and to be recognized as a well-adjusted model to improve maternal health outcomes in a country where tackling maternal mortality constitutes a public health top-priority (EDSM 2021 provide an estimate of 424 maternal death for 100 000 live birth, among the highest in the world).

Our findings reveal the shocking burden of SV inflicted on adolescent girls as soon as they enter reproductive age, with concerns specific to pregnancy risk and loss of virginity likely resulting in their over-representation among the USPEC caseload. Yet IPV cannot be understood as an adult-only matter, especially in a country where child marriage is persistent. Child marriage fuels the risk of IPV: isolated from their families, disempowered and highly dependent on their spouse and spouse's family economically, emotionally and regarding access to information, married children and adolescents face multiple forms of victimization. In addition to this concern regarding adolescent girls, 21.91% (n = 778) of victims had not yet reached the age of 12, which cannot be explained by FGM cases only (n = 24). This highlights the need to assess GBV beyond the reproductive age bracket (15–49 years old) generally used in research and policy briefs [4, 6, 7], and limits the opportunity for comparison with other studies.

Though adolescent pregnancy appeared to be the most prevalent complication associated with GBV investigated in this study, mental health repercussions are likely very frequent as well. Standard care at the USPEC encompasses psychological first aid and an initial assessment to establish the need for specialist attention. Yet no psychiatrists nor trained psychologists are available outside of the capital, and in Nouakchott psychiatric attention is limited to cases requiring hospital admission or psychoactive treatment, leaving psychosocial assistants in charge of nearly all emergency and follow-up care. Due to their limited capacity to provide for a sound assessment of victims' psychological status, mental health outcomes were not investigated. However, Mauritania has recently engaged in a national strategy addressing mental health as an indissociable part of health, offering a way forward in structuring and staffing mental healthcare. Physical lesions, such as wounds or trauma, were not infrequent but offer limited generalization potential to understand the landscape of GBV. On the one hand, the tertiary-level nature of the USPEC may lead to an over-representation of GBV victims with severe lesions such as fractures or suturable wounds; on the other hand, IPV and FGM, which often cause physical lesions, are likely under-represented in the cohort, especially compared to recent population census of these forms of violence [21]. We make an argument below that many, but not all renounce to healthcare: some seek medical attention elsewhere, and data about these consultations can improve our understanding of GBV in Mauritania.

FGM victims may, dramatically and owing to the illegal character of the practice, never be presented to medical facilities. They also may seek care in other settings, either because life-threatening complications such as hemorrhage, infections and anuria would be directly referred from emergency units to operating theatres, or because they experience later complications, while the USPEC–owing to financial and operational constraints–limit care delivered to emergency conditions. The latter hypothesis is substantiated by a recent retrospective case-study of surgical repairs performed after FGM in Nouakchott [26], which analyzes the socio-demographic, socio-cultural and clinical characteristics of 42 patients who received surgical treatment after FGM complications, in two hospitals in Nouakchott between January 1st, 2018 and December 31st, 2020. Among this cohort, patients were generally under 5 years-old (66.6%) and coming from rural areas (61.9%). They presented with dysuria (40.47%), clitoral swelling (30.9%), or urinary complications such as incontinence or acute retention (21.42%) that had appeared within 12 months (29%), 12 to 24 months (31%) or more after the FGM. These complications were often related to synechia of the labia minora (50%), and treatment accordingly consisted in surgically releasing the synechiae. Later complications, which also include hematocolpos occurring after menarche, dyspareunia or impossibility to have sexual relations, dystocic childbirth, chronic pelvic pain, lesions of the perineum, and vesico-vaginal and vesico-anal fistulas, may be more prevalent than urgent ones. While the EDSM 2021 presents FGM as a mere antecedent, more research is needed to document FGM complications in

order to increase evidence-based advocacy to eradicate the practice. Surgical options for late complications, such as desinfibulation, deobstruction of the urethral meatus, resection of fibrosis in the case of vaginal synechiae, and progressive vaginal dilatation, could also be integrated to the comprehensive care delivered through the USPEC. Again, more focus is needed on proactive screening during antenatal care, to assess for the risk of dystocic labor as a consequence of FGM; improving preemptive, therapeutic surgical options could reinforce the USPEC function within a strategy to prevent maternal morbidity and mortality. Women and girls facing mental health issues after FGM such as post-traumatic stress disorders, anxiety and depression, psychosomatic disorders such as enuresis, changes in body image, could also benefit from the psychological attention delivered in the USPEC.

Regarding IPV, the EDSM 2021 states that 10% of women aged 15–49 have experienced physical violence, and almost one in five women has experienced emotional, physical and/or sexual violence committed by a husband, which suggest that IPV is also under-represented in this caseload. Though only a qualitative, socio-anthropological approach could establish whether this is due to a certain pattern of violence–IPV–that is under-represented, or a certain population–married, adult women–that is less likely to seek healthcare, or both, a few assumptions can be made about plausible limited recourse to healthcare. First, women and girls who were subjected to violence but did not need emergency medical care, or the ones not willing to press charges and hence not having a medico-legal certificate established, may perceive less benefit in consulting after an episode of IPV than after an episode of SV–here again, sensitization programs appear crucial in awareness-raising. Like for FGM, these patients may also consult elsewhere, as suggested by a 6-months retrospective study from the forensic medicine department of the Nouakchott Military Hospital [27], which documents 76 cases of physical violence perpetrated on women between February 1st and July 31st, 2023, in Nouakchott area. In this cohort, mean age was 44.6 years–compared to the present study where only 21.30% were 18 years old or above–, and all victims presented with lesions from blunt (92%) or sharp weapon, as well as burns. But more worrying, victims of IPV may face more barriers to access healthcare than other GBV-victims due to fear of retaliation, disempowerment in decision-making regarding their own health, lack of economic means to pay for transport, deprivation of liberty, and so on, all of which underlie IPV.

Even if data on marital status were often lacking, which limits the scope for generalization for the GBV landscape in Mauritania, it can be inferred from the high proportion of minors that married women were under-represented. This is generally explained by the fear of major social prejudice associated with sexual intercourse occurring outside of wedlock for married and/or adult women, even in the context of rape: they are at risk of being prosecuted under criminalization of *zina*, and punished by incarceration. More than their male counterparts, they also risk dramatic social consequences such as rejection by the husband, family and stepfamily. A widespread belief is that married women are *unrapable* [inviolable], as if rape could occur in a virgin only: such barriers should be more thoroughly investigated through a socio-anthropological approach.

While married women were likely under-represented, this is also likely the case of male victims. The USPEC being set-up within maternity wards could deter men and boys from seeking care therein. Certain documented patterns of SV against men may further hinder their capacity to seek care, such as rape and sexual abuse occurring in prisons and in the army, or amongst men who have sex with men, as the practice is criminalized in Mauritania. Other specific patterns of GBV against male include ill-treatment of male external genitalia, from abuse to torture as part of sexual games and hazing among adolescents, and rape and incest in the family and school settings. The present study is consistent with the latter, with a higher

proportion of SV occurring in the school setting amongst boys than girls, and with a known, recidivist perpetrator, though the small caseload does not allow to generalize these findings.

In this study, the over-representation of children, compared to adults, is coherent with the current trend of addressing concurrently violence against women and girls and violence again children [28]. Though this did not represent a substantial caseload, cases of physical violence on children (n = 9), perpetrated by the parents or the entourage as a punishment for alleged sexual misconduct (see Fig 2), constitutes abuse and maltreatment that add to the list of children rights violation investigated more in-depth, such as FGM and child marriage. These forms of violence imposed from an early age, prolonged through the aforementioned "reproductive coercion" that deprive young adults from their rights and autonomy, are to be understood as features of an intergenerational continuum of violence, which produce similar–if not cumulative–detrimental social norms and consequences on individual mental, physical, sexual and reproductive health.

## Study limits

Limited to the setting of 6 USPEC countrywide, this study exhibits obvious selection biases. It could not capture the true nature and scope of GBV in Mauritania, especially amongst the most vulnerable populations, as it excluded those who did not seek care or sought care from non-hospital sources. Stigma and fear often lead victims of GBV not to disclose their experience; a lack of knowledge of available services and their potential benefits, or limited resources to reach them, also affect how likely they are to seek care. Since most victims were referred by the police and to a hospital, the caseload may over-represent the most severe cases, i.e. rapes in children, rapes resulting in pregnancies, or physical violence associated with severe injury.

Data from this study relied on the analysis of an "artisanal" excel database, developed incrementally over the years and by non-specialists. The database likely contained inaccuracies in data-entry, due to inconsistent data coding formats and manual entry by health workers with limited training for the task. Data from 2017, consisting of 41 cases seen in CHME, were excluded due to too many erroneous entries as the tools were just developed and the teams just being trained. Additionally, the database's source material consisted in medical records, filled with a varying degree of quality depending on the context (e.g. victims presenting with a medical emergency or intense distress), and the professionals' language literacy (French or Arabic being equally represented among the USPEC teams as preferred language, and often neither being their native tongue). Incomplete records hence limit the possibility to fully characterize the demographical and medical parameters investigated. An additional layer of complexity stems from producing consistent case definitions in the field of GBV. Even well-defined forms of violence allow for interpretations during data entry regarding which "type" should be recorded, since most victims may have faced a continuum of violence rather than an isolated form. As previously discussed, some "types" also spontaneously arose from the practice and are context-specific. While these are highly informative of the Mauritian-context, they limit the scope for generalization of this study's findings. The same limitations apply to the qualification of the perpetrator: most USPEC teams used not to recognize as an intimate partner a non-married one, and hence rather categorized them under another category, such as relative (e.g. cousin) or entourage (boyfriend). This produced discrepancies between the proportion of IPV victims and the proportion of perpetrators referred to as intimate partner.

As previously addressed, the investigation of complications associated with GBV was limited, particularly regarding mental health issues, but not only. Future, prospective studies should assess USPEC's efficacy to deliver prophylaxis by investigating the occurrence of infections with HIV, hepatitis B or other sexually transmitted infections among victims of sexual

violence. Pregnancy outcomes–e.g. live birth, neonatal weight, maternal complications and death–and follow-up for the children born from these pregnancies–e.g. development, nutrition–should also be prospectively investigated to ground the USPEC model as a pathway for improved neonatal, maternal and reproductive health outcomes for teenage pregnancy.

These limitations reveal a need to enhance data quality, achievable through the development of an improved, standardized tool and training for data-entry. To further build the case for the USPEC model's effectiveness also requires to investigate victims' health outcomes beyond initial consultation. From these learnings, a digitalized database was developed with a user-friendly, streamlined interface for both initial and follow-up cases–the latter integrating parameters such post-exposure prophylaxis and pregnancy outcomes. As of January 2024, this new database has been deployed, paving the way for future, more robust and generalizable research on GBV.

## Conclusion

Despite these limitations, this study provides valuable insights into the GBV landscape in Mauritania. Shedding a light on the characteristics of the victims seeking care in the USPEC, and the nature and patterns of violence they were subjected to, it outlines some of the social dynamics detrimental to women, and especially to girls, that are involved in GBV. A pattern emerges: that of under-aged girls, raped by perpetrators from their entourage and in their own home, seeking care through police referral, and with a concern to prevent or declare a pregnancy. This study also outlines a continuum of violence from childhood to adulthood, depriving children and women from their rights, including their right to mental, physical, sexual and reproductive health Yet power dynamics may also affect who is not seeking healthcare–victims of physical and psychological violence perpetrated by their partner, and prevented from accessing healthcare; married women, who may even be criminalized for disclosing being raped; young and baby girls subjected to an illegal genital mutilation, and borne to secrecy; boys, for whom it is imposed by social norms to never cry and complain, and always consent to sex. More research is needed to better understand the practical and sociocultural bottlenecks that limit access to comprehensive care and support for victims.

Findings also suggest that the USPEC model adequately responds to both the victims' and the health system's need to address GBV, though future research should investigate whether healthcare delivered in the USPEC is associated with improved mental and physical health outcomes at individual level. The comprehensive model of the USPEC, addressing both medical and socio-legal needs, constitutes a keystone to help victims seek care, justice and protection. Owing to the specific characteristics of GBV victims attending to the USPEC, these units contribute to recognize the prejudice of forced pregnancy in unmarried women, teenagers and children, and tackle maternal morbidity and mortality by providing free, comprehensive and quality care to pregnant adolescents.

The implementation of the USPEC model, as a targeted intervention, also offers a way forward in how a program led by an international organization can successfully integrate within the health system, by taking the financial risk of setting up an advanced model of care with a clear path towards local adoption and scale-up. The Ministry of Health and many regional hospitals are currently eager to disseminate this model, and MdM strongly encourages such appropriation through the dissemination of its medical protocols, clinical and organizational tools, and the training of medical, paramedical and psychosocial professionals nationwide. In that sense, this study also tackles the misconception that the GBV response and discourse are necessarily undermined by the conservative fringe of an Islamic Republic; instead, Mauritania here exemplifies leadership in innovative practice and health policy.

## Supporting information

**S1 File. Dataset.**
(XLSX)

**S2 File. Dataset_description.**
(PDF)

## Acknowledgments

The authors wish to thank Dr Pierre-Yves Lablanche, senior data scientist, for his contribution to data cleaning, extraction and visualization. We extend our appreciation to Dr María del Carmen Díez Hernandez and María Jesús Girona for their review, inputs and suggestions. Our deepest appreciation goes to the USPEC teams; among them, special thanks go to Neyba Ndiaye, Gansiry Camara, Diouma Sow, Fatou Gueye and Fatimata Ba, who tirelessly ensured complex case management, healthcare quality, data entry, and, thanks to their contextual knowledge and deep insight, data analysis.

## Author Contributions

**Conceptualization:** Clairanne Bost, Mouhamedou Diagana.

**Data curation:** Clairanne Bost.

**Formal analysis:** Clairanne Bost.

**Investigation:** Houssein Lebkem.

**Methodology:** Clairanne Bost, Mouhamedou Diagana.

**Writing – original draft:** Clairanne Bost.

**Writing – review & editing:** Clairanne Bost, Mouhamedou Diagana, Houssein Lebkem.

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
