## [Decision Letter · Decision Letter 0]

9 May 2024

PGPH-D-24-00134

Gender violence care in Mauritania: Experience and caseload of six specialized hospital units (2018-2023)

Dear Dr. Bost,

Thank you for submitting your manuscript to PLOS Global Public Health. After careful consideration, we feel that it has merit but does not fully meet PLOS Global Public Health’s publication criteria as it currently stands. Therefore, we invite you to submit a revised version of the manuscript that addresses the points raised during the review process.

We look forward to receiving your revised manuscript.

Kind regards,

Muthusamy Sivakami

Academic Editor

Journal Requirements:

Additional Editor Comments (if provided):

Thank you for submitting the paper. Our reviewers have asked for revisions to the paper which are very valid. Kindly revise the paper based on their review and submit at the earliest. This will allow us to move to next stage of this paper.

I look forward to receiving the paper at the earliest.

Reviewers' comments:

Reviewer's Responses to Questions

**Comments to the Author**

1. Does this manuscript meet PLOS Global Public Health’s publication criteria? Is the manuscript technically sound, and do the data support the conclusions? The manuscript must describe methodologically and ethically rigorous research with conclusions that are appropriately drawn based on the data presented.

Reviewer #1: Yes

Reviewer #2: Yes

2. Has the statistical analysis been performed appropriately and rigorously?

Reviewer #1: Yes

Reviewer #2: Yes

3. Have the authors made all data underlying the findings in their manuscript fully available (please refer to the Data Availability Statement at the start of the manuscript PDF file)?

Reviewer #1: Yes

Reviewer #2: Yes

4. Is the manuscript presented in an intelligible fashion and written in standard English?

Reviewer #1: Yes

Reviewer #2: Yes

5. Review Comments to the Author

Reviewer #1: Thank you so much for the opportunity to review this interesting paper on gender-based violence. The authors have done an incredible job. I have few comments regarding this paper, which I think, if I addressed, will make this paper stronger.

1. Under the inclusion criteria in the Methods section, the authors indicated that there is no age restriction, and they have provided detailed information regarding adolescent pregnancies and child marriages. I am wondering if detailed information could also be provided on child abuse—perhaps including child abuse/maltreatment as a key term as well.

2. According to the authors, approximately 78% were girls under 18 years old. I was expecting a more detailed discussion on why this is the case in this context. I believe the authors should further elaborate, such as what insights can be gained from this?

3. There is need for copyediting the whole article for grammatical errors. For example, line 148 the authors wrote ‘were included all cases’ instead of ‘we included’. There are several areas with these areas, I think the authors should take some time to really copyedit the manuscript.

4. The authors should also ensure consistency in the use of acronyms such as Gender-Based Violence (GBV), Intimate Partner Violence (IPV), and World Health Organization (WHO). We use the full terms the first time we mention them and use the acronyms throughout the manuscript.

I wish the authors the best of luck!!

Reviewer #2: This study analyzes data from six USPEC units in Mauritania to investigate the characteristics of gender-based violence (GBV) against adolescent females. It highlights the challenges they face in accessing care. The research offers valuable insights into GBV in Mauritania and strengthens the case for the USPEC model as a potential tool for combating GBV and improving victim healthcare. However, the reviewer report should delve deeper into the study's methodology and limitations to fully assess its contribution to advancing gender equality efforts.

Here are some areas for potential improvement:

1. Briefly mention the research question or objective within the introduction.

2. Consider adding a sentence about the sample size (number of cases) at the beginning of the "Data collection and material" section (line 165), for better context.

3. Briefly mention the statistical methods used for data analysis in the "Data analysis" section (line 176). For example, you could state that "descriptive statistics" were used to summarize the data or that "chi-square tests" were used to compare proportions between groups, as the case may be.

4. The discussion section thoroughly analyzes results, positions them within existing research, acknowledges limitations, and proposes future research directions. The discussion could be strengthened by including a future research direction that explores the long-term impact of the USPEC model on victims' health and well-being. This could involve following up with participants after a certain period to assess the model's effectiveness in supporting their recovery and overall well-being.

5. The discussion transparently acknowledges limitations (selection bias, data entry errors), explains their impact (over-representing severity), and details potential consequences for the findings. Briefly mention recommendations or mitigation strategies for the identified limitations. For example, you could state that future studies might benefit from standardized data collection tools and improved training for data entry personnel.

6. The study's reliance on police referrals for data collection likely underrepresents the true prevalence of gender-based violence, particularly domestic violence and Female Genital Mutilation (FGM). Future research could address this limitation by incorporating data from additional sources, such as shelters, NGOs, and community health services.

6. PLOS authors have the option to publish the peer review history of their article (what does this mean?). If published, this will include your full peer review and any attached files.

**Do you want your identity to be public for this peer review?** For information about this choice, including consent withdrawal, please see our Privacy Policy.

Reviewer #1: No

Reviewer #2: No

---

## [Editor Report · Decision Letter 1]

5 Jun 2024

Gender-Based Violence care in Mauritania: Experience and caseload of six specialized hospital units (2018-2023)

PGPH-D-24-00134R1

Dear Mrs Bost,

We are pleased to inform you that your manuscript 'Gender-Based Violence care in Mauritania: Experience and caseload of six specialized hospital units (2018-2023)' has been provisionally accepted for publication in PLOS Global Public Health.

Best regards,

Muthusamy Sivakami

Academic Editor

Thank you for revising your paper based on the comments received from our reviewers.